# "Because I said so." – Collection and evaluation of parenting phrases in German-speaking samples

Erika Kljucak ⓘD, Nourat N. Alazza ⓘD, Pia Hemme, Louisa Kulke ⓘD*

Department of Developmental Psychology with Educational Psychology, University of Bremen, Bremen, Germany

* kulke@uni-bremen.de

## Abstract

Verbal communication is a key aspect of parenting behaviour. However, phrases frequently used in parenting situations (e.g., "Stop crying", "I love you") have yet to receive focused attention in the research literature. This set of two online questionnaire studies investigated the use of parenting phrases within a German-speaking sample. The first study compiled an inventory of common parenting phrases, contextualized them within parenting style research on the dimensions of warmth and control, and examined their associations with self-esteem. A total of 309 phrases were collected, of which the 84 most frequently mentioned were further analysed in Study 2, which examined their ratings within a communication theory perspective and investigated socio-demographic influences on participants' evaluations. Participants reported hearing more negative and fewer positive phrases from their caregivers than they reported using towards their own children. Similarities between the frequency of negative or positive phrases heard during participants' upbringing and the frequency of phrases from those categories used with their own children could not be found. Neither parental warmth and control nor exposure to self-esteem-related phrases were significantly associated with self-esteem. Age differences in experienced parenting phrases and parenting phrase usage were observed, as younger participants reported hearing more positive phrases and using fewer negative phrases than older participants. In Study 2, the ratings of the parenting phrases were related to participants' age and experienced parenting during childhood, but no association was found with parental status. Exploratory results from both studies align with the Relational Framing Theory, suggesting it may serve as a useful addition to traditional parenting style theories in interpreting caregiver-child communication. These studies provide a foundation for investigating parenting phrases, thereby offering a novel perspective on parental communication – a crucial factor in shaping child development and family dynamics.

**Data availability statement:** All data and analysis files are available from the OSF database (accession link: https://osf.io/wh72j/files).

**Funding:** The author(s) received no specific funding for this work.

## Introduction

Phrases like "Why can't you be more like your sister?", "I'm proud of you" or "You do as I say" are commonly heard during childhood. Repeated exposure to such messages, both positive and negative, can shape a person's self-perception and beliefs, potentially influencing individuals well into adulthood. Despite substantial evidence linking parent-child communication to key developmental outcomes [1–4], it is often assessed only through broad items (e.g., "[My parent] Made statements that communicated to me that I was a unique, valuable human being." ([5], p. 329) or isolated expressions (e.g., parents' use of the phrase "You're being difficult on purpose" ([6], p. 56) as an indicator of parental invalidation). To our knowledge, only one study has systematically examined the specific parenting phrases within a Bulgarian context [7]. Despite their prevalence, little is known about how these phrases are perceived and the mechanisms through which they influence a person's development. This project is the first to systematically investigate parenting phrases within the German-speaking population. For this purpose, parenting phrases are operationalized as any phrase, saying or advice regularly used by caregivers in parenting situations. They can cover a wide range of topics and can convey messages that can be perceived in a positive or negative way.

This definition overlaps partially with the concept of "parental messages", which Levkova defines as "statements addressed in different situations personally to the child as 'lessons' or 'live wisdom', as something that is self-explanatory (life axioms) and as statements exchanged between adults in the presence of the child" ([7], p. 52). While our definition aligns with the notion of verbally conveyed meanings occurring across diverse parenting contexts, the present construct is broader in that it includes not only explicit lessons or axioms but also recurring everyday utterances. These include, for instance, expressions of affection, disciplinary remarks, or critical comments.

Importantly, parenting phrases do not constitute an independent parenting behaviour, separate from other parenting constructs. Rather, they reflect the verbal manifestation of established parent-child interaction patterns. For example, they may serve as vehicles of verbal discipline [8], contain elements of verbal aggression [9], or function as expressions of psychological control [10]. Similarly, they may contribute to children's emotional socialization by communicating norms regarding emotional expression and regulation. However, emotional socialization also encompasses non-verbal processes such as parental modelling and responsiveness [11], which extend beyond the scope of the present construct.

Thus, parenting phrases are best understood as the linguistic component embedded within broader parenting behaviours. Although this broad operationalization entails conceptual overlap with related constructs, it allows for a systematic examination of parental language as a distinct and measurable aspect of parenting. Future research may further differentiate context-specific categories of parenting phrases. However, at this early stage of investigation, a broad definition seems useful to capture the full range of parental verbal expression.

Beyond these conceptual considerations, it is also important to acknowledge the cultural context in which such phrases emerge. While there are likely some

cross-cultural similarities in the contexts in which these phrases occur and the functions they serve, the specific wording and content can be expected to differ across cultures, therefore reflecting linguistic characteristics unique to the German language and parenting culture.

Two online questionnaire studies were conducted to gain a deeper understanding of this underexplored topic. Study 1 employed a mixed-methods design to collect a list of parenting phrases familiar to participants, either from their own upbringing or from their experience with their own children. To our knowledge, this is the first comprehensive collection of such expressions in the German language, providing a foundation for future research in this area. To situate these phrases within the context of existing parenting research and examine possible societal and intergenerational changes in their usage, a subsequent quantitative analysis investigated their associations with warmth, control, self-esteem and age, as well as intergenerational continuity. However, individuals can also differ in the way they interpret these statements. For some, phrases like "Take care of yourself" could convey affection and concern for a child's well-being, while others might perceive them as overprotective and restricting the child's sense of freedom. This is particularly relevant, as children's developmental outcomes are more strongly linked to their subjective perception of parenting than to their parents' assessments [12] or the assessments of objective observers [13]. To investigate differential perceptions of parenting phrases, Study 2 used a selection of the most common phrases identified in Study 1, asking participants to rate them along the parenting dimensions. It further explored associations between demographic variables and these evaluations. Findings were interpreted through the lens of Relational Framing Theory [14], offering valuable insights into the intersection of parenting and communication theories, an area that, as pointed out by Estlein [15], remains underexplored.

## Study 1

### Research questions and hypotheses

**2.1.1. Collection of phrases and correspondence to parenting dimensions.** The first study aimed to identify what types of parenting phrases are commonly used. The main goal was to generate, categorise, and qualitatively analyse these phrases in order to develop a comprehensive inventory of parenting phrases. For this purpose, participants were first asked to freely write down parenting phrases that they had previously heard. Subsequently they rated their familiarity with a list of parenting phrases compiled by the authors. As a next step of Study 1, quantitative analyses were conducted to further deepen the knowledge on these phrases. To embed the collected parenting phrases within existing parenting research, a widely recognized classification of parenting behaviour was used. According to Baumrind [16] and Maccoby and Martin [17], parenting behaviour can be described along the two dimensions "parental warmth" (or "parental responsiveness") and "parental control" (or "parental demandingness"). Detailed definitions as presented to participants are available in the Procedure section. Parenting phrases were categorised according to the extremes of these dimensions ("high warmth", "low warmth", "high control" and "low control"). It was hypothesized that there is a correlation between the frequency of hearing phrases from these categories and the assessment of caregivers' behaviour on the corresponding dimensions (H6a-d). (Note that the wording from the preregistration was imprecise and is further explained in Supplement S1 Text.) Evidence for these associations would indicate that parenting phrases can be understood as a verbal component of these broader parenting constructs.

**2.1.2. Intergenerational changes and continuities in parenting phrase usage.** When asking participants about their caregivers' parenting behaviour and, if applicable, about their own behaviour towards their children, it is reasonable to assume that their perceptions are not objective. Instead, they might be subject to various biases. For instance, people tend to interpret and remember events in a way that serves a self-enhancing or self-protective function, portraying themselves more favourably [18]. Some evidence supporting this idea in the context of the evaluation of parenting behaviour was provided by Campbell and Gilmore [19]. They found that individuals described their own behaviour as less authoritarian – a parenting style characterised by low warmth and high control and typically associated with a variety

of negative outcomes [20–22] – than they recalled their parents' behaviour to have been. This effect likely applies to the reporting of parenting phrases as well. We therefore hypothesized that more negative phrases will be remembered from childhood than are stated to be used in people's own parenting practices (H1). Likewise, we expect participants to claim to use more positive phrases towards their children than they remember hearing in their own childhood (H2). However, we cannot determine whether any observed differences reflect a self-favouring bias or actual changes in parenting behaviour due to societal shifts and a possible effort to be better parents than participants' own parents were. For example, Eschner [20] conducted a qualitative analysis of German parenting guides from 1945 to 2015. The study delineated a development from post-war parenting priorities focused on obedience and submission to a greater emphasis on autonomy, affection, and fostering a positive parent-child relationship. Similar trends were observed in interviews across three generations of parents [21]. Quantitative research on the topic further supported these findings [22,23]. To identify potential changes in parental communication, we analysed associations between participants' year of birth and the phrases. Specifically, it was expected that older participants would report having heard (H5a) and used (H5b) more negative phrases than younger participants. Given today's increased accessibility of parenting information [24], the harmful consequences of negative parental language might be more apparent, leading to a more critical perspective on such phrases in younger generations. On the other hand, research on parenting styles has also consistently documented their transmission across generations [19,24,25]. Based on this work, we hypothesized that similar patterns would emerge for the use of parenting phrases. Specifically, the more negative, respectively positive, phrases participants recall from their own childhood, the more likely they are to use negative, respectively positive, phrases in the parenting of their own children (H3 & H4). This approach situates parenting phrases within the framework of intergenerationally transmitted parenting behaviour.

### 2.1.3. Parenting phrases in relation to self-esteem.

Lastly, a key focus of parenting research is drawing conclusions for children's developmental outcomes. Initial evidence of a connection between parenting phrases and such outcomes was provided by Levkova [7], who found that aggressive adolescents were more likely to have been exposed to predominantly negative parental phrases, whereas their non-aggressive peers reported hearing primarily positive ones. Similarly, perceived parental messages were shown to be related to children's perceptions of themselves, the world, and their future [2]. This led us to hypothesize that the use of specific parenting phrases could be linked to children's self-esteem. Parenting style literature has consistently demonstrated a significant association between parenting behaviour and children's self-esteem. Specifically, research indicates that parenting characterised by high warmth and high control is positively associated with self-esteem [26–29]. In contrast, parenting with low warmth and high control, often referred to as authoritarian parenting style, was found to be negatively correlated with self-esteem [26,27,29–31]. Similarly, Kernis et al. [32] demonstrated critical and psychologically controlling parental behaviour to be associated with low and unstable self-esteem. Parenting that is low in both warmth and control is also negatively associated with self-esteem [29,30]. However, findings on parenting that is characterized by high warmth but low control are less consistent. Some studies showed no significant relationship [26]. Others suggest that this form of parenting shows equally strong [30], in fact sometimes even stronger, positive associations with self-esteem than those observed for parenting high in warmth and control [33]. This highlights the primary role of parental warmth, both with and without parental control, in fostering children's self-esteem. However, longitudinal studies also suggest that parental control negatively affects self-esteem [34,35]. Therefore, we examined the associations between the parenting dimensions warmth (H8a) and control (H8b) and self-esteem, hypothesizing that parental warmth would be positively correlated with self-esteem, whereas parental control would show a negative correlation. Additionally, we investigated the relationships between exposure to parenting phrases categorized as positive (self-esteem-enhancing) and negative (self-esteem-inhibiting), as well as their ratio, with self-esteem (H7a-c). In doing so, we aimed to replicate previous findings on associations between parenting dimensions and self-esteem, and to examine whether parenting phrases show similar links.

## Method

### Sample

Study 1 was approved by the local ethics committee of the University of Bremen (Reference number 2023–27) and was preregistered prior to data collection ([36], https://doi.org/10.17605/OSF.IO/F2Q4Y). It was conducted between 19 and 20 December 2023. Participants were recruited via mailing lists and online platforms. Students from the University of Bremen could obtain course credit for their participation. Since the qualitative generation of parenting phrases was the main objective of this study, a priori considerations on theoretical saturation in qualitative knowledge generation (see [37]) were made. These considerations resulted in a desired sample size of at least 60 participants, including a minimum of 20 participants with children, as this subsample was necessary to examine hypotheses 1–4 and 5b. Data collection continued until these targets were met. The inclusion criteria required participants to be proficient in the German language and at least 18 years old. A total of 82 participants completed the online questionnaire, including 21 parents. In this sample, 71 participants identified as female, 9 as male and 2 as nonbinary. Their year of birth ranged from 1961 to 2005 ($M = 1995$, $SD = 9.19$). 58 participants were native German speakers. However, 7 participants had to be excluded from quantitative analysis due to an error in displaying the questionnaire, and another participant was excluded because they had not answered the control question correctly and exhibited implausible answering behaviour. This, in consequence, resulted in a final sample of 74 complete data sets for quantitative analysis. For the qualitative analysis of parenting phrases, the whole sample of 82 participants was used.

### Phrase selection and categorisation process

Alongside the phrases generated by participants, the authors had previously compiled and categorised parenting phrases based on their prior knowledge and experience. These were subsequently presented to participants as part of the study. To categorise the phases, a process of expert validation was used. First, five independent raters classified the phrases into the categories "high warmth", "low warmth", "high control", "low control", "positive/self-esteem-enhancing" or "negative/self-esteem-inhibiting". Secondly, the five ratings for each phrase were aggregated, with a phrase considered to belong to a category if at least four raters agreed. This allowed for a single phrase to be classified into more than one category (e.g., both "high warmth" and "low control") and resulted in the 64 categorized parenting phrases presented in the study. The number of phrases in each category is presented in the table below (Table 1). A list of the phrases with their assigned categories is provided in the supplements (S1 Table). Fleiss' $\kappa$ was used to determine interrater reliability. All dimensions showed moderate to almost perfect agreement, except "low control", which reached only fair agreement, indicating difficulties in rating phrases on this dimension. Detailed values, including standard errors and confidence intervals, are depicted in Table 1.

Table 1. Categorisation of parenting phrases presented in Study 1: Number of phrases per category and interrater reliability (Fleiss' $\kappa$).

| Category | High Warmth | Low Warmth | High Control | Low Control | Self-esteem-enhancing (positive) | Self-esteem-inhibiting (negative) |
|---|---|---|---|---|---|---|
| Number of phrases | 19 | 36 | 24 | 3 | 19 | 25 |
| Fleiss' $\kappa$ | 0.82 | 0.62 | 0.46 | 0.24 | 0.68 | 0.46 |
| SE | 0.07 | 0.10 | 0.11 | 0.12 | 0.09 | 0.11 |
| 95%-CI | [0.68, 0.96] | [0.43, 0.82] | [0.24, 0.68] | [-0.00, 0.48] | [0.50, 0.86] | [0.24, 0.68] |

Fleiss' $\kappa$ reflects interrater agreement for category assignment across five coders. SE = Standard error of $\kappa$; 95%-CI = 95% confidence interval for $\kappa$. Because individual phrases could be assigned to more than one category, the number of phrases per category is not mutually exclusive.

## Procedure

Participants filled out an online questionnaire that was developed and presented using the software SoSciSurvey [38]. Participants provided written consent to the terms of participation by clicking a mandatory field that was required to proceed with the questionnaire. Subsequently, they were asked for sociodemographic information (i.e., gender, year of birth, country of origin, native language) as well as information on their family (i.e., number and gender of siblings and children, and information on caregivers). After that, participants were provided with definitions of parental warmth and control, which were created based on literature research. They were asked to rate the parenting of their primary and, if applicable, secondary caregiver on those dimensions using a slider. The definition given for the dimension "Emotional warmth" was: "The caregiver's behaviour is perceived as loving, supportive, praising, and comforting. The caregiver shows affection, appreciation, care, and pleasure when interacting with the child. The caregiver is emotionally available, positively involved in the child's activities, sincere and caring." [39–43]. The second dimension, "Control", was defined as: "The caregiver has expectations and demands of the child. The child should obey and fulfil the caregiver's demands. The caregiver has power and sets firm rules. If the child violates these rules, they are punished by the caregiver." [41,42]. In the following section of the questionnaire, participants were asked to list as many parenting phrases as they could think of in an open-ended question format. Additionally, participants rated their familiarity with each phrase using the following options: "I am familiar with this phrase", "I have heard others say this phrase to others", "This was said to me by my caregivers", and, for participants with children, "I use this phrase with my children". Additionally, if participants selected the option "I cannot think of anything", they were given a prompt with two examples of parenting phrases and then asked to try listing phrases again. In a next step, participants rated the familiarity of the 64 parenting phrases categorised in advance. The items were presented in randomized order. A control question was included to test for attentiveness. After that, participants were given another opportunity to list any additional parenting phrases they might have come up with. They were also instructed to imagine a positive parenting situation from their own childhood (and, if applicable, from their own parenting) and recall the phrases caregivers used in that situation. Participants were then asked to repeat this process for a negative parenting situation. Subsequently, participants completed the revised Rosenberg self-esteem scale [44]. Lastly, participants had the option to comment on the questionnaire.

## Qualitative analysis

Parenting phrases that were repeatedly mentioned or had very similar meanings were summarized and categorized. The frequency with which phrases were mentioned served as an indicator of their familiarity.

## Statistical analyses

For hypothesis testing, one-sided, dependent t-tests (H1 and H2) and correlation analyses (H3-H8) were used. (Please see S1 Text for details of this correction of the preregistration.) To facilitate comparisons between phrase categories, the number of phrases heard or said from each category was divided by the total number of phrases in that specific category. This yielded relative frequencies, ranging from 0 to 1, that allowed for a meaningful comparison between categories. For all statistical analyses, an alpha level of 0.05 was used to determine significance. All statistical analyses were conducted using RStudio (Version 2024.09.1 + 394, [45]).

## Results

### Qualitative analysis

In total, 866 parenting phrases were mentioned by the participants. Redundant phrases and phrases with similar meanings were grouped together, resulting in 309 parenting phrases, which can be accessed in the supplements (S2 Table). The phrases were sorted into the existing categories. Two additional categories were identified: the first being "rules of etiquette" (e.g., "Wash your hands before eating") and the second being "idioms" (e.g., "No pain, no gain"). 15 phrases

could not be assigned to any of the categories. The number of phrases in each category can be viewed in Table 2. Consistent with the findings of Levkova [7], the majority of collected phrases were categorised as negative. However, a substantial number of positive phrases were also identified, several of which ranked among the most frequently mentioned. These include: "I am proud of you" (39 times), and "You can do it" (37 times). Others reflected more controlling attitudes, for example, "As long as you live under my roof, you'll do as I say" (German: "So lange du deine Füße unter meinen Tisch stellst, tust du was ich sage", 20 times). The examples range from expressions of emotional support and encouragement to more authority-driven statements, illustrating the diverse nature of parental communication.

## Quantitative analysis

### Prerequisite testing

To test prerequisites for the dependent t-tests of hypotheses 1 and 2, the Shapiro-Wilk test was used to assess the normality of difference scores, and boxplots were used to check for outliers. The Shapiro-Wilk test was nonsignificant, indicating that the normality requirement was met. No outliers were found for the relevant variables. For the other hypotheses, correlation analyses were calculated. Again, Shapiro-Wilk tests were conducted for all variables included in the hypotheses to assess the assumption of normality. For each confirmatory hypothesis, at least one variable showed a significant deviation from normality ($p<.05$). Consequently, Spearman rank correlations were used for all analyses. Outliers were identified using boxplots and subsequently excluded from the analyses. The results differed depending on whether the outliers were included or excluded, with some effects reaching statistical significance only in one of the conditions. For transparency, both sets of results, and all prerequisite analyses are reported in the supplements (S1 Analysis).

### Hypothesis testing

Table 3 presents means, standard deviations and ranges for all key variables included in the analyses.

**Hypothesis 1.** We predicted that participants would report hearing more negative phrases in childhood than they report using with their own children. Consistent with this prediction, a paired-samples t-test showed a significant difference between the frequency of negative phrases remembered and negative phrases used in participants' own parenting practices, $t(19) = 2.04$, $p=.028$, 95% CI [0.02, ∞], $d=0.66$, $BF=1.28$. Based on the phrases presented to the participants, they reported hearing, on average 0.13 more negative phrases from their caregivers than they reported saying to their own children (Fig 1A).

**Table 2. Categorisation of phrases collected in Study 1: Number of parenting phrases per category.**

| Category | Number of phrases |
| --- | --- |
| Positive valence | 72 |
| Negative valence | 123 |
| High warmth | 76 |
| Low warmth | 137 |
| High control | 123 |
| Low control | 31 |
| Rules of etiquette | 24 |
| Idioms | 60 |
| Not assignable | 15 |

*Note that categories are not mutually exclusive, individual phrases may be classified under more than one category.*

**Table 3. Descriptive statistics of variables included in the analyses of Study 1.**

| Variable | M | SD | Range (sample) | Range (scale) |
|---|---|---|---|---|
| NP heard from caregivers (complete sample) | 0.36 | 0.21 | 0-0.8 | 0-1 |
| NP heard from caregivers (subsample with children) | 0.34 | 0.20 | 0-0.76 | 0-1 |
| NP said to children | 0.21 | 0.19 | 0-0.6 | 0-1 |
| PP heard from caregivers (complete sample) | 0.49 | 0.30 | 0-0.89 | 0-1 |
| PP heard from caregivers (subsample with children) | 0.29 | 0.27 | 0-0.79 | 0-1 |
| PP said to children | 0.72 | 0.13 | 0.47-0.89 | 0-1 |
| Year of birth (complete sample) | 1994.64 | 9.19 | 1961-2005 | / |
| Year of birth (subsample with children) | 1982.2 | 6.88 | 1961-1992 | / |
| HC phrases heard from caregivers (complete sample) | 0.44 | 0.20 | 0-0.92 | 0-1 |
| LC phrases heard from caregivers (complete sample) | 0.30 | 0.30 | 0-1 | 0-1 |
| HW phrases heard from caregivers (complete sample) | 0.53 | 0.30 | 0-0.95 | 0-1 |
| LW phrases heard from caregivers (complete sample) | 0.38 | 0.19 | 0-0.81 | 0-1 |
| Control dimension (primary caregiver) | 43.39 | 31.20 | 1-101 | 1-101 |
| Control dimension (secondary caregiver) | 41.24 | 30.41 | 1-101 | 1-101 |
| Warmth dimension (primary caregiver) | 78.43 | 27.84 | 1-101 | 1-101 |
| Warmth dimension (secondary caregiver) | 68.89 | 31.14 | 1-101 | 1-101 |
| Self-esteem | 32.04 | 5.25 | 19-40 | 10-40 |

*M = mean; SD = standard deviation; Range (sample) = observed minimum and maximum values in the sample; Range (scale) = theoretical minimum and maximum values of the scale. Abbreviations for phrase categories: NP = negative phrases, PP = positive phrases, HC = high control, LC = low control, HW = high warmth, LW = low warmth*

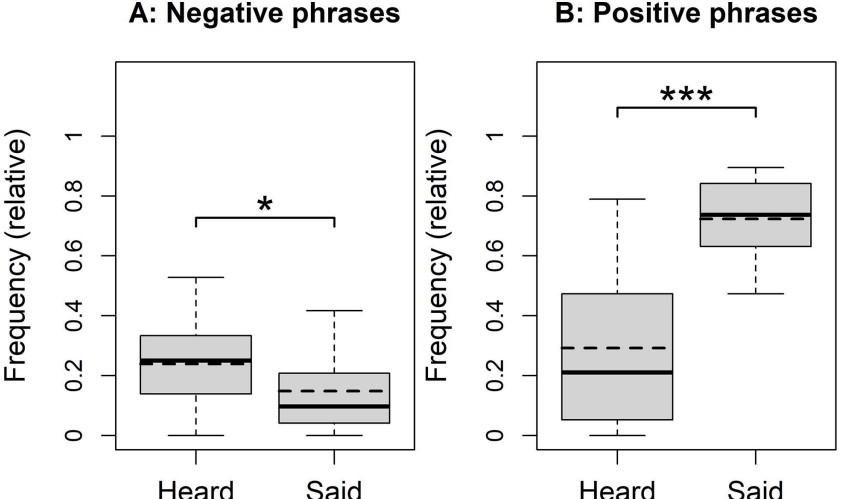

**Fig 1. Boxplots showing the relative number of negative and positive phrases heard and said.** The dashed line represents the mean, and the solid line indicates the median of the data. Significance levels are indicated by asterisks: p < .05 (*), p < .001 (***).

**Hypothesis 2**. Consistent with H2, a paired-samples t-test showed a significant difference between the number of positive phrases remembered from participants' own childhood and phrases used towards their children, $t(19) = 7.18$, $p < .001$, 95% CI [0.33, ∞], $d = 1.98$, $BF = 2.2 \times 10^4$. On average, participants reported saying 0.43 more positive phrases towards their children than they reported hearing from their caregivers (Fig 1B).

**Hypothesis 3**. H3 predicted a positive association between the frequency of negative phrases remembered from participants' childhood and the frequency of negative phrases used towards their own children. This hypothesis was not supported, as the correlation was not significant, $\rho = -.02$, $p = .934$, 95% CI [−0.46, 0.43], $BF = 0.48$.

**Hypothesis 4**. Contrary to prediction, no significant correlation was found between the frequency of positive phrases remembered from participants' childhood and the frequency of positive phrases they reported using towards their own children, $\rho = .18$, $p = .455$, 95% CI [−0.29, 0.58], $BF = 0.59$.

**Hypothesis 5a**. While we predicted participants' with an earlier year of birth to have heard more negative phrases from their caregivers, compared to participants' with a later year of birth, the observed correlation was not significant, $\rho = -.04$, $p = .707$, 95% CI [−0.27, 0.18], $BF = 0.29$. (Fig 2A)

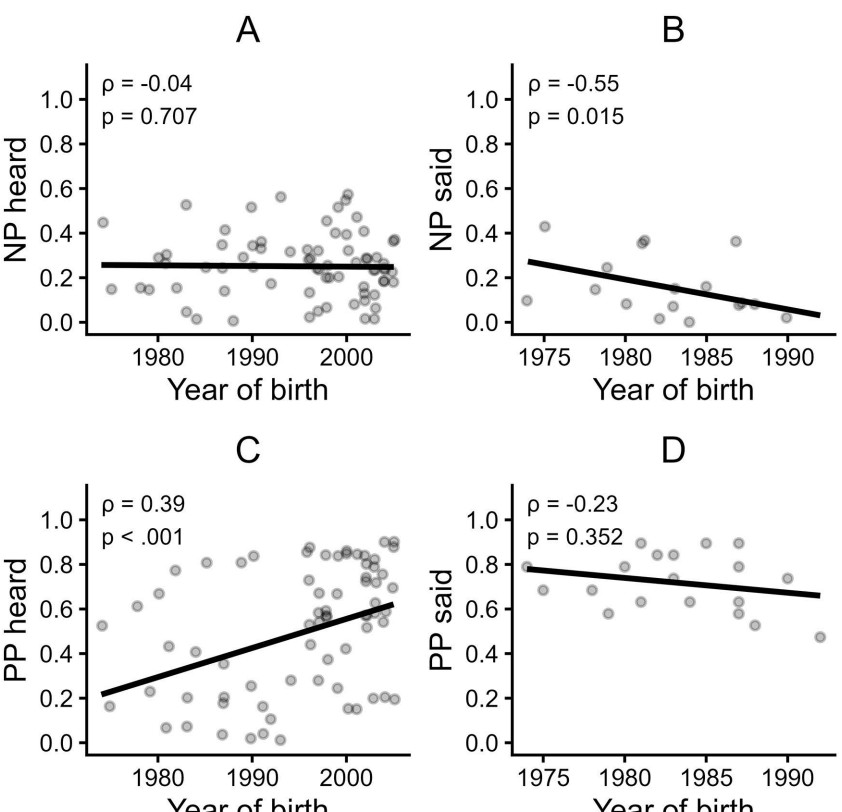

**Fig 2. Scatter plots of participants' year of birth versus relative frequencies of phrases heard and said.** Spearman correlation coefficients (ρ) and p-values are indicated on each plot. Note that for the phrases said, the smaller subsample of participants with children (n = 20) was used, while for the phrases heard, the full sample with complete quantitative data (n = 74) was used. Abbreviations: NP = negative phrases, PP = positive phrases.

 

**Hypothesis 5b.** In line with H5b, a significant negative correlation was found between participants' year of birth and the frequency of negative phrases they reported saying to their children, $\rho = .55$, $p = .015$, 95% CI [−0.80, −0.13], BF = 4.84. (Fig 2B)

**Hypothesis 6a.** Consistent with the hypothesis, the frequency of phrases categorised as "high control" heard by caregivers was significantly positively correlated with the parenting dimension control for the primary caregiver ($\rho = .39$, $p < .001$, 95% CI [0.17, 0.57], $BF = 45.36$), as well as for the secondary caregiver ($\rho = .4$, $p = .002$, 95% CI [0.16, 0.59], $BF = 6.18$).

**Hypothesis 6b.** We predicted a negative association between the frequency of phrases categorised as "low control" heard from caregivers and the parenting dimension control. However, this hypothesis was not supported, as the correlation was not significant for either the primary caregiver ($\rho = .18$, $p = .132$, 95% CI [−0.05, 0.39], $BF = 0.76$) or for the secondary caregiver ($\rho = .21$, $p = .103$, 95% CI [−0.04, 0.20], $BF = 7.33$).

**Hypothesis 6c.** As predicted, the frequency of phrases categorised as "high warmth" heard by a participant's caregivers was significantly positively correlated with the parenting dimension warmth, both for the primary caregiver ($\rho = .35$, $p = .002$, 95% CI [0.13, 0.54], $BF = 20.35$) and the secondary caregiver ($\rho = .3$, $p = .015$, 95% CI [0.06, 0.51], $BF = 0.91$).

**Hypothesis 6d.** Contrary to prediction, no significant correlation was found between the frequency of phrases categorised as "low warmth" heard by a participant's caregivers and the parenting dimension warmth, neither for the primary caregiver ($\rho = −.193$, $p = .114$, 95% CI [−0.41, 0.05], $BF = 0.88$), nor for the secondary caregiver ($\rho = −.02$, $p = .866$, 95% CI [−0.27, 0.23], $BF = 0.3$).

**Hypothesis 7a.** While we predicted self-esteem to be positively associated with the frequency of parenting phrases with positive valence (categorised as "self-esteem-enhancing"), no significant correlation was observed, $\rho = .11$, $p = .363$, 95% CI [−0.12, 0.33], $BF = 0.39$.

**Hypothesis 7b.** Contrary to our hypothesis, no significant correlation was found between the frequency of parenting phrases with negative valence (categorised as "self-esteem-inhibiting") and self-esteem, $\rho = −.13$, $p = .268$, 95% CI [−0.35, 0.10], $BF = 0.47$.

**Hypothesis 7c.** The results did not support H7c, as the relation of positive to negative phrases heard from caregivers was not found to be significantly correlated with self-esteem ($\rho = .24$, $p = .057$, 95% CI [−0.01, 0.45], $BF = 1.51$).

**Hypothesis 8a.** H8a predicted parental warmth to be positively associated with self-esteem. However, the hypothesis was not supported. The parenting dimension of warmth was not significantly correlated with self-esteem, neither for the primary caregiver ($\rho = .19$, $p = .104$, 95% CI [−0.04, 0.41], $BF = 0.92$), nor for the secondary caregiver ($\rho = −.1$, $p = .427$, 95% CI [−0.33, 0.15], $BF = 0.38$).

**Hypothesis 8b.** While we predicted parental control to be negatively associated with self-esteem, the results did not support this hypothesis. We found no significant correlation between self-esteem and the parenting dimension of control neither for the primary caregiver ($\rho = −.17$, $p = .143$, 95% CI [−0.39, 0.06], $BF = 0.72$), nor for the secondary caregiver ($\rho = −.09$, $p = .477$, 95% CI [−0.33, 0.16], $BF = 0.31$).

### Exploratory analyses

There was a significant positive correlation between a participant's year of birth and the frequency of positive phrases they reported hearing by their caregiver, $\rho = .39$, $p < .001$, 95% CI [0.18, 0.57], $BF = 60.56$ (Fig 2C). No significant correlation was found between the participants' year of birth and the frequency of saying positive phrases towards their children, $\rho = −.28$, $p = .241$, 95% CI [−0.66, 0.16], $BF = 0.68$ (Fig 2D). A significant negative correlation was found between the frequency of heard phrases categorised as "low control" and the warmth dimension for the primary caregiver ($\rho = −.32$, $p = .007$, 95% CI [−0.32, −0.09], $BF = 8.14$), but not for the secondary caregiver ($\rho = −0.1$, $p = .376$, 95% CI [−0.34, 0.14], $BF = 0.27$). A significant positive correlation was found between the frequency of heard phrases categorised as "low warmth" and the control dimension for both the primary caregiver ($\rho = .27$, $p = .023$, 95% CI [0.04, 0.48], $BF = 3.04$), and the secondary caregiver ($\rho = .32$, $p = .014$, 95% CI [0.07, 0.53], $BF = 3.94$).

## Discussion Study 1

The first study's main objective was to collect phrases frequently used in the parenting context. In total, 309 parenting phrases were collected and categorised, providing a useful inventory for future research on parental communication (S2 Table).

The study further aimed to connect this emerging research field with the well-established theory of parenting styles, which conceptualizes parenting behaviour along the dimensions of parental warmth (or responsiveness) and parental control (or demandingness) [16,17]. The following correlations reflect the observed patterns within the current sample: Phrases categorised as "high control" were found to be positively correlated with the assessment of parenting behaviour on the control dimension, both for the primary and for the secondary caregiver. The same was found for the association of phrases categorised as high in warmth and the parenting dimension warmth. Contrary to the hypotheses, phrases categorised as low in control were not significantly correlated with control, and the same was observed for "low warmth" phrases and the "warmth" dimension. These findings raise the possibility that the dimensional approach of traditional parenting style theories [17], originally developed to describe broader interaction patterns, may not translate directly to the level of individual parenting phrases. In addition to these considerations, exploratory analyses revealed a negative correlation between "low control" phrases and the warmth dimension and a positive correlation between "low warmth" phrases and the control dimension. What may seem surprising at first, given the absence of a significant association between these phrases and their corresponding dimension, can be tentatively interpreted through the lens of the Relational Framing Theory [14]. This theory states that interactional messages can be interpreted through only one of two interpretational frames at a time, which correspond in content to the dimensions of warmth (affiliation-disaffiliation frame) and control (dominance-submissiveness frame). From this perspective, it is conceivable that phrases low in warmth may be processed primarily within a dominance-submissiveness frame, whereas phrases low in control may activate the interpretation through an affiliation-disaffiliation frame. These exploratively observed patterns invite consideration of relational framing processes as a possible explanatory perspective. Relational Framing Theory may therefore offer a conceptual framework to examine parental communication at the level of individual phrases. Importantly, this interpretation does not constitute empirical support for Relational Framing Theory. Rather, it should be regarded as hypothesis-generating rather than confirmatory, providing a theoretical basis that is explored in greater depth in Study 2. Overall, the effects observed in the current study were weaker for secondary caregivers, suggesting a stronger influence of primary caregivers on perceived parenting phrase usage, likely due to their more prominent role in day-to-day parenting interaction.

Notably, some conceptual concerns arise regarding the "low control" category, which was characterized by low inter-rater reliability and comprised only three phrases. While the limited number of phrases is theoretically justifiable – given that low control, particularly in the context of neglectful parenting, may involve "a lack of interaction" ([46], p. 2164) – the combination of sparse item representation and low agreement among raters constitutes a limitation of the present study. In addition, the "high control" and "self-esteem inhibiting" categories reached only moderate interrater agreement, indicating some ambiguity in the rating of these characteristics. Consequently, results related to these categories should be interpreted with caution, as lower reliability may reduce the robustness of observed associations. In response to these concerns, Study 2 employed a dimensional evaluation of the phrases, enabling a more differentiated examination of their perceived characteristics.

Another aim of the study was to explore potential continuities and discontinuities in the use of parenting phrases across generations. While previous research has demonstrated transgenerational continuity in parenting behaviour [19,24,25], the results of the present study do not suggest similar patterns for the use of parenting phrases. Specifically, in this sample no significant correlations were found between the frequency of phrases heard in childhood and their use with participants' own children, for either negative or positive phrases. These results may indicate that findings on the inter-generational transmission of parenting styles cannot simply be transferred to parenting phrases. One possible explanation for this divergence is that research on intergenerational transmission has primarily focussed on broader behavioural and attachment patterns [24,25,47]. Verbal statements may be more consciously processed and context-dependent

than global interaction patterns and thus potentially less susceptible to replication across generations. Moreover, the assessment of both childhood experiences and participants' own parenting behaviour relied on retrospective self-report. Reconstructive memory processes and social desirability biases may have limited the detection of potential continuity effects. A further methodological explanation concerns statistical power. Post hoc sensitivity analyses indicated that, for these hypotheses, the study was only sufficiently powered ($1 - \beta = .80$, $\alpha = .05$) to detect correlations of $r \geq .552$. Thus, the present findings cannot rule out the presence of smaller effects, which may have gone undetected in this sample. Future studies should therefore aim to recruit larger samples of parents to examine intergenerational transmission.

Regarding potential discontinuities, comparisons between the frequency of positive and negative phrases participants reported hearing in their own childhood and those they reported using with their own children revealed significant differences: Participants in the current sample reported using significantly fewer negative phrases and significantly more positive phrases with their children than they recalled hearing throughout their own childhood. However, from these results it cannot be concluded that this equates to actual parenting practice. While some research suggests a shift in parenting values toward more affectionate and supportive approaches [20–23], which could be reflected in an increased use of positive and decreased use of negative phrases, an alternative explanation is equally plausible – namely, that participants may exhibit a bias favouring their own parenting practices over those of their caregivers. To further investigate which of these interpretations applies, a follow-up study was conducted involving caregiver–child dyads to capture both perspectives on parenting behaviour (preregistration: [48], https://osf.io/dhv2e/). Findings from this study may be presented in future work.

As research by Ecarius [21] and Eschner [20] indicates, parenting values have undergone significant changes over the last century. In line with this, analyses of participants' year of birth and their reported experience and use of parenting phrases yielded notable results: While the current study observed no significant correlation between participants' year of birth and the frequency of negative phrases heard during childhood, participants born earlier reported using significantly more negative phrases toward their own children. This age-related pattern may suggest cohort differences in parenting practices. Alternatively, social desirability effects might have influenced the responses, with younger participants perhaps having greater awareness of the impact of negative language and therefore being more reluctant to report its use. This could also explain the lack of an effect for negative phrases heard, which might be less affected by social desirability than participants' own behaviour. Another possible explanation relates to the age of participants' children. Younger participants are likely to have younger children, which may influence the type of language they use. Since this study did not assess the age of participants' children, it remains unclear whether this variable plays a role. Additionally, exploratory analyses revealed a significant positive correlation between year of birth and the frequency of positive phrases heard during childhood. These results indicate that in the current study younger participants reported hearing positive phrases more frequently. However, no such correlation was found for their use of positive phrases. One possible explanation may lie in a confounding of age and parenthood within this study: While participants' years of birth ranged from 1974 to 2005, the subsample of parents was considerably on the older side of that range, born between 1974 and 1992 (excluding one outlier born in 1961 from both samples). A visual inspection of the distribution of positive phrase frequency (Fig 2) suggests a possible rise in the popularity of positive parenting phrases beginning in the mid-1990s and continuing into the early 2000s – approximately at the time our parent subsample would have begun raising children. Consistent with this observation, the parent subsample on average reported a notably high frequency of positive phrase use, with low variability. This likely limited the potential to detect meaningful associations. While we cannot determine with certainty whether this finding reflects an actual increase in the phrase usage or a reporting bias among parents, the corresponding result regarding positive phrases heard (with younger participants from this sample reporting having heard positive phrases more frequently) appears to support the former explanation. To better understand these possible trends, future studies should include a more balanced sample of both younger and older parents and take into account their children's year of birth.

Lastly, no significant association between self-esteem and parenting was found for this sample, neither for parenting phrases nor for the parenting dimensions. This is somewhat surprising, given the extensive body of literature supporting

a connection between parenting behaviour and self-esteem development [26–35]. Several methodological considerations might account for this divergence from previous findings. One possibility is that characteristics of the current sample contributed to the absence of significant associations. While the variability in self-esteem scores was similar to that reported in earlier research [44], the overall level of self-esteem in the sample was comparatively high. This may have led to a ceiling effect, reducing the ability to detect meaningful associations with parenting variables, particularly those linked to lower self-esteem. Additionally, a sensitivity analysis indicated that the present sample allowed detection of effects of $r ≥ .313$ ($1 − β = .80$, $α = .05$). While the sample size in Study 1 was determined to be sufficient for the qualitative aims of the study, relations between self-esteem and parenting may require larger samples, as a meta-analysis on the associations between parenting and self-esteem reported predominantly small to moderate effect sizes [49]. Smaller effects may therefore have remained undetected in the current sample. Some concerns also arise for the operationalisation of the warmth and control dimensions. In this study, participants received brief descriptions of each dimension and were asked to evaluate their caregivers' general behaviour throughout their whole childhood using a single item per construct. This approach does not adequately capture the multifaceted nature of parenting behaviour and overlooks the possibility of certain developmental stages in which parenting phrases might be more strongly linked to self-esteem. Future studies should employ more comprehensive and developmentally sensitive measures of parenting to gain a deeper understanding of these dynamics. Lastly, a possible conceptual explanation may concern the use of global self-esteem as an outcome measure. Global self-esteem may be too broad and distal to capture associations with the relatively specific use of parenting phrases. More general parental behaviours may be more strongly related to global self-esteem, whereas parenting phrases may be more closely linked to proximal and specific developmental outcomes. Accordingly, future research could focus on outcomes such as self-criticism, attachment security and satisfaction with the parent-child relationship, which have been shown to relate to parental verbal communication [50–52]. Contingent self-esteem may also be especially relevant, as it reflects a more specific, conditional aspect of the self-concept and may relate to parents' verbal implications of conditional worth. Research could also focus on examining associations with domain-specific self-esteem, which may be particularly sensitive to verbal expressions that target evaluative content within the respective domain. In addition, the phrase corpus generated in the present study could inform intervention-based research. For example, Infante [53] and Wilson et al. [54] propose strategies that can help parents to reduce verbal aggression towards their children. Such approaches could be adapted to the context of parenting phrases to examine whether changes in parental language are associated with changes in theoretically aligned outcomes.

Study 1 provides valuable first insights into the largely unexplored topic of parenting phrases, presenting a substantive collection of common phrases for future research. Given the complexity and potential impact of this form of communication, more comprehensive research is essential. Study 2 extended this work by investigating how individuals perceive these phrases, contributing to a deeper understanding of the subjective interpretation of parental communication.

## Study 2

### Research questions and hypotheses

Building on the findings of Study 1, Study 2 aimed to validate the initial categorisation of the collected parenting phrases by having participants rate them on the dimensions warmth, control, and self-esteem enhancement, as well as to explore demographic differences in how these phrases are evaluated.

### Parenting phrases through the lens of the Relational Framing Theory

Study 1 related the phrases to the parenting dimensions described by Baumrind [16]. An alternative theoretical framework, Relational Framing Theory (RFT) by McLaren [14], suggests that messages in interactions can be interpreted through either a dominance-submissiveness frame or an affiliation-disaffiliation frame. The theory posits that both frames cannot

be activated simultaneously, as they inhibit each other. Exploratory findings from Study 1 indicate that this theory may be well-suited for application to parenting phrases. In Study 2, the ratings of parenting phrases were analysed to explore which theoretical framework better accounts for the emerging concept of parenting phrases.

### Sociodemographic differences in phrase evaluation

Another objective of this study was to examine how ratings of the parenting phrases vary across individuals, as they may apply different evaluative standards based on their own experiences. To our knowledge, no prior research has directly examined how individuals' perceptions of their upbringing influence their evaluations of parenting behaviour. The perception of a person's own upbringing might serve as a reference frame to compare the phrases to. H1 predicted that participants' perceptions of their own upbringing (regarding warmth, control, and self-esteem enhancement) would be inversely related to their ratings of the parenting phrases on the corresponding dimensions. Similar to Study 1, Study 2 also investigated potential age-related differences in the recalled exposure to, and additionally in the rating of parenting phrases. Prior research suggests a historical shift in parenting values [19,21,22]. Study 1 provided initial evidence of such a trend, with younger participants reporting having heard more positive phrases throughout their childhood and using fewer negative phrases when interacting with their children. However, age-related differences concerning phrases related to parental warmth and control have yet to be examined. To address this, Study 2 investigated how age correlates with the frequency of parenting phrases heard, hypothesizing that younger participants would report hearing phrases classified as "high warmth" more frequently (H3a), while older participants were expected to report hearing phrases classified as "low warmth" (H3b) and "high control" (H3c) more often. Phrases from the "low control" category were excluded due to low interrater reliability and sparse representation in Study 1. This assumed shift in experienced parenting may also shape how parenting phrases are evaluated, raising expectations for emotional warmth and self-esteem enhancement, while fostering a more critical view of parental control. Based on this idea, we hypothesized that younger participants would rate the phrases as less emotionally warm (H2a), more controlling (H2b), and less self-esteem-enhancing (H2c). Finally, parenthood was expected to influence perceptions of parental communication. For example, a review by Bačić [55] found that parents and non-parents differ significantly in their attitudes towards corporal punishment. It was expected that they might also evaluate other aspects of parental behaviour differently, which would be reflected in their ratings of the parenting phrases (H4).

## Method

### Sample

After receiving approval from the local ethics committee of the University of Bremen (Reference number 2023−27) and preregistering the study ([56], https://doi.org/10.17605/OSF.IO/9WFVB), data collection took place from February 28, 2024 until March 17, 2024. Participants were required to be of legal age and to have been raised in a German-speaking household. The desired sample size for Study 2 was determined based on two power analyses calculated using G*Power (Power = .8, $\alpha$ = .05) [57]. Power considerations for Hypothesis 1 were based on a review of intergenerational continuity of parenting behaviour [25], indicating an average effect size of $r = .3$, which led to a desired sample size of 67. For Hypotheses 2 and 3, studies on intergenerational differences in parenting [22,23] were reviewed, suggesting an expected effect size of r = .25 and thereby an optimal sample size of 97 participants. Due to the lack of prior research on the potential effects of parenthood on the evaluation of the relevant parenting behaviours, no power analysis could be conducted for Hypothesis 4. Consequently, a conservative minimum sample size of 97 participants was determined. Recruitment took place via contacts, social media, and announcements in kindergartens and sports groups. A total of 185 people participated in the study, of whom 77 were excluded for not completing the questionnaire, and two for not answering the attention control question correctly, resulting in a final sample of 106 participants. Of those participants, 24 identified as male

and 82 as female. For this sample, the mean age was 43 years with a standard deviation of $SD = 18.1$ and a range of 18–77 years. 60.4% of participants reported having children, whose ages ranged from 0 to 56 years.

### Phrase selection process

Since Study 1 had generated over 300 parenting phrases, the phrases included in Study 2 were preselected in the following manner: Only phrases that were mentioned by at least three participants of Study 1 were included, and similar phrases were summarised to prevent redundancies. This resulted in 84 parenting phrases that were included in the study and can be viewed in the supplements (S3 Table).

### Procedure

The study was conducted using an online questionnaire designed and presented on the platform SoSciSurvey [58]. After giving written consent to the participation requirements by clicking a mandatory field that was required to proceed with the questionnaire, participants were asked demographic questions regarding their age, gender, occupation, and number and ages of their children. After that, participants were provided the definitions of the parenting style dimensions warmth and control (corresponding to the ones that had already been used in Study 1), as well as the following definition of self-esteem enhancement: "Self-esteem-enhancing parenting helps to ensure that a child develops a sense of self-worth and perceives themselves as a valuable person. They can appreciate their own successes, but also recognise their mistakes. If, on the contrary, an upbringing leads to a child not developing self-esteem and considering itself unworthy, insufficient or deficient in other respects, this upbringing inhibits self-esteem" (based on [59]). If necessary, participants could look up these definitions while answering the questionnaire. They were asked to rate the parenting they had experienced throughout their childhood on those three dimensions, using a five-point Likert scale ranging from "low warmth" to "high warmth", from "low control" to "high control", and from "inhibits self-esteem" to "enhances self-esteem". After rating the general parenting behaviour of their parents, participants were presented with the 84 parenting phrases and were asked to rate them on the same Likert scale as well as on another five-point scale ranging from "never heard this in my upbringing" to "heard this in my upbringing very often". In this part of the questionnaire, a control question was included to test whether participants were answering attentively. At the end, participants were given the opportunity to provide feedback on the questionnaire.

### Statistical analyses

For hypothesis testing, Spearman correlation analyses (H1-H3) and two-sided independent t-tests (H4) were used. An alpha level of 0.05 was applied throughout the analyses to determine significance. All statistical analysis was conducted using RStudio (Version 2024.09.1 + 394, [45]).

## Results

### Categorisation of parenting phrases and descriptive analysis

The phrases were categorised according to the parenting styles described by Maccoby and Martin [17], resulting in the four possible combinations "high warmth/ high control", "high warmth/ low control", "low warmth/ high control", and "low warmth/ low control". Additionally, phrases were categorised as "self-esteem-enhancing" and "self-esteem-inhibiting". A phrase was categorised as "low" on any of the dimensions (i.e., warmth, control, self-esteem enhancement, and frequency of hearing the phrase), if the mean rating was lower than 3 on the 5-point scale. Correspondingly, a phrase was categorised as "high" on a dimension if the mean rating was higher than 3. Frequencies of the phrases by category are shown in Table 4. It is notable that most phrases are either high in control and low in warmth (57.1%) or high in warmth and low in control (31.0%). Combinations of high or low ratings on both dimensions are rare.

**Table 4. Categorisation of parenting phrases in Study 2: Number of phrases and percentages by category.**

| Category | Number of phrases | Percentage |
|---|---|---|
| Self-esteem-enhancing | 29 | 34.5% |
| Self-esteem-inhibiting | 55 | 65.5% |
| High warmth/ high control | 4 | 4.8% |
| High warmth/ low control | 26 | 31.0% |
| Low warmth/ high control | 48 | 57.1% |
| Low warmth/ low control | 6 | 7.1% |

*Note that phrases each fall into one of the two self-esteem-relevant categories and one of the parenting dimension combinations. Percentages and number of phrases are calculated separately for the self-esteem and parenting dimension categories, with each classification set summing to the total number of phrases.*

Mean ratings and standard deviations for each phrase are provided in the supplements (S3 Table).

### Prerequisite testing

Outliers were identified using boxplots and removed to ensure the robustness of the analyses. Notably, the inclusion or exclusion of these outliers influenced the statistical significance of some results. Results from both versions of the analyses, along with all prerequisite analyses, are presented in the supplements for comparison (S2 Analysis). For the variables used in the independent t-tests (Hypothesis 4), Levene's test was performed to assess the homogeneity of variances. This assumption was violated for the control dimension. Consequently, Welch's test was used as an alternative. The robustness of the t-tests, or Welch's test respectively, against deviations from normality was assumed due to the sample size for this hypothesis.

### Hypothesis testing

Table 5 presents means, standard deviations and ranges for all key variables included in the analyses.

**Hypothesis 1a.** H1a predicted experienced warmth in participants' upbringing to be negatively related to phrase ratings on the warmth dimension. The hypothesis was not supported, as no significant correlation was found, $\rho = .18$, $p = .07$, 95% CI [−0.01, 0.35], $BF = 1.08$. (Fig 3A)

**Hypothesis 1b.** While we predicted a negative association between the rating of a person's own upbringing on the control dimension and the control rating of parenting phrases, no significant correlation was found, $\rho = .07$, $p = .509$, 95% CI [−0.13, 0.26], $BF = 0.28$. (Fig 3B)

**Hypothesis 1c.** The correlation between the self-esteem enhancement rating of one's own upbringing and the parenting phrases was in the opposite direction than hypothesized: A significant positive correlation was found, $\rho = .33$, $p < .001$, 95% CI [0.15, 0.49], $BF = 59.46$. (Fig 3C).

**Hypothesis 2a.** As hypothesized, a significant positive correlation was found between participants' age and the rating of the phrases on the warmth dimension, $\rho = .25$, $p = .012$, 95% CI [0.04, 0.40], $BF = 4.63$ (Fig 4A).

**Hypothesis 2b.** H2b predicted younger participants to rate the phrases as less controlling. Contrary to our expectation, no significant correlation was found between participants' age and the rating of the phrases on the control dimension, $\rho = −.17$, $p = .084$, 95% CI [−0.35, 0.02], $BF = 0.94$ (Fig 4B).

**Hypothesis 2c.** The hypothesis predicted that younger participants would rate the phrases as less self-esteem enhancing. However, no significant correlation was found between participants' age and the rating of the phrases on the self-esteem-enhancement dimension, $\rho = .11$, $p = .268$, 95% CI [−0.08, 0.39], $BF = 0.40$ (Fig 4C).

**Table 5. Means and standard deviations of variables included in the analyses of Study 2.**

| Variable | M | SD | Range (sample) |
|---|---|---|---|
| Experienced warmth | 3.91 | .96 | 1-5 |
| Experienced control | 3.00 | 1.12 | 1-5 |
| Experienced self-esteem enhancement | 3.55 | 1.23 | 1-5 |
| Phrase rating warmth | 2.79 | 1.20 | 1.07-4.86 |
| Phrase rating control | 3.22 | .67 | 1.95-4.33 |
| Phrase rating self-esteem enhancement | 2.83 | 1.08 | 1.09-4.81 |
| Frequency of hearing phrases categorised as high in warmth | 2.89 | .64 | 1.07-4.07 |
| Frequency of hearing phrases categorised as low in warmth | 2.46 | .63 | 1.31-4.37 |
| Frequency of hearing phrases categorised as high in control | 2.51 | .63 | 1.35-4.40 |
| Age | 43.44 | 18.07 | 18-77 |

*All variables (experienced parenting, phrase dimensions and frequency) were rated on a five-point Likert scale (1 = low, 5 = high). The table reports the mean (M), standard deviation (SD), and observed range (minimum and maximum) in the sample.*

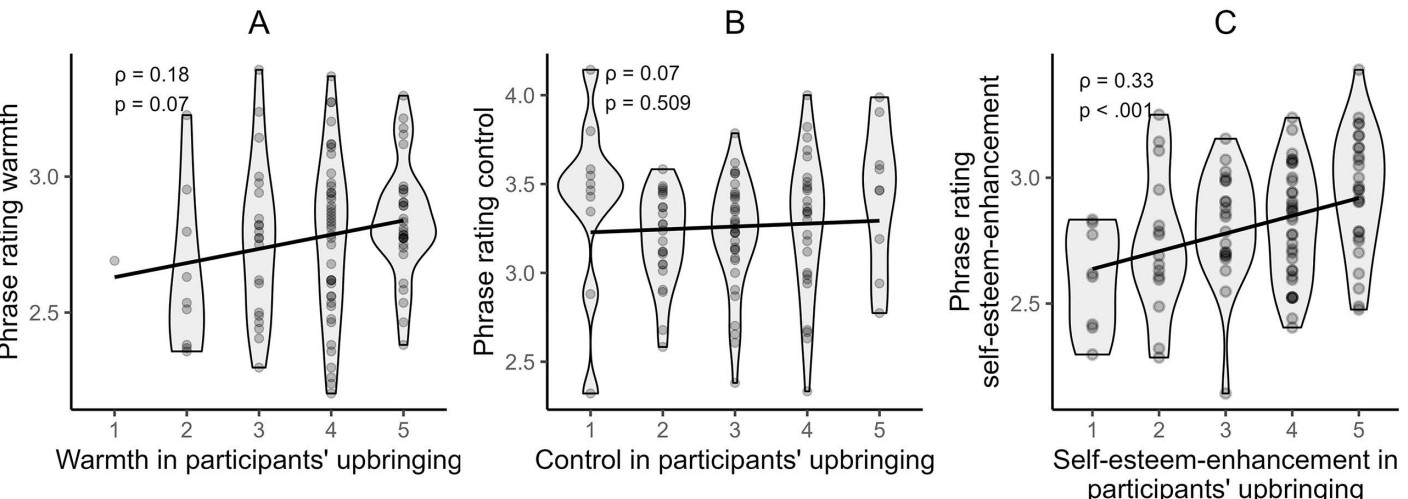

**Fig 3. Violin plots illustrating the associations between phrase ratings and the parenting behaviour across the parenting dimensions.** *Spearman correlation coefficients (ρ) and p-values are indicated on each plot.*

**Hypothesis 3a.** We hypothesized that younger participants would report hearing phrases classified as "high warmth" more frequently. However, no significant correlation was found between participants' age and the reported frequency of hearing phrases categorised as high in warmth from their caregivers, $\rho$ = .00, $p$ = .983, 95% CI [−0.19, 0.19], $BF$ = 0.22.
**Hypothesis 3b.** H3b predicted older participants to report hearing "low control" phrases more often. In contrast to our expectation, no significant correlation was found between participants' age and the reported frequency of hearing phrases categorised as low in warmth from their caregivers, $\rho$ = −.04, $p$ = .654, 95% CI [−0.24, 0.15], $BF$ = 0.25.
**Hypothesis 3c.** We predicted that older participants would report hearing "high control" phrases more frequently. The result does not support this hypothesis, as no significant correlation was found between participants' age and the reported frequency of hearing phrases categorised as high in control from their caregivers, $\rho$ = −.04, $p$ = .715, 95% CI [−0.23, 0.16], $BF$ = 0.24.

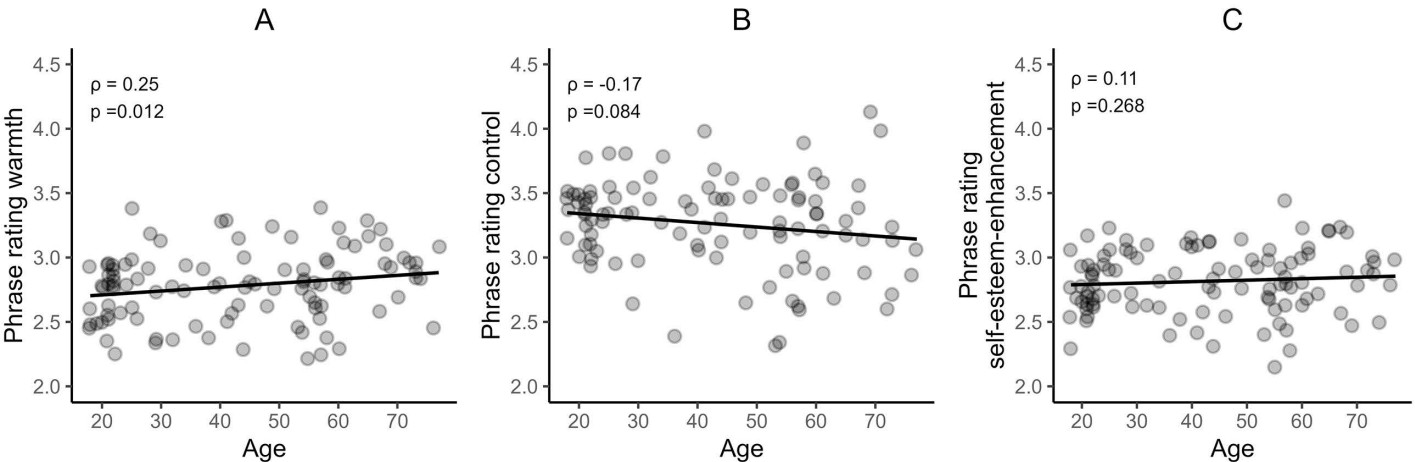

**Fig 4. Scatterplots illustrating the relationship between age and phrase ratings on the dimensions warmth, control, and self-esteem enhancement.** *Spearman correlation coefficients (ρ) and p-values are indicated on each plot.*

**Hypothesis 4a.** Inconsistent with the hypothesis, no significant difference was found for participants with and participants without children concerning their ratings of phrases on the warmth dimension, $t(103) = 0.35$, $p = .731$, 95% CI [−0.09, 0.13], $d = 0.07$, $BF = 0.22$.

**Hypothesis 4b.** Contrary to this hypothesis, the Welsh-test showed no significant difference for participants with and participants without children for their ratings of phrases on the control dimension, $t(99.505) = −1.39$, $p = .166$, 95% CI [−0.09, 0.13], $d = −0.26$, $BF = 0.45$.

**Hypothesis 4c.** In contrast to our expectation, no significant difference was found for participants with and participants without children concerning their ratings of phrases on the self-esteem-enhancement dimension, $t(103) = −0.33$, $p = .74$, 95% CI [−0.12, 0.08] $d = −0.07$, $BF = 0.22$.

### Exploratory analyses

An explorative analysis of the relationship between the dimensions of warmth, control and self-esteem enhancement revealed a significant positive correlation between warmth and self-esteem enhancement at both the parenting behaviour ($\rho = .66$, $p < .001$, 95% CI [0.54, 0.76], $BF = 3.7 \times 10^{11}$) and the parenting phrases ($\rho = .97$, $p < .001$, 95% CI [0.96, 0.98], $BF = 5.05$) levels. Warmth and control were not significantly correlated at the dimension level ($\rho = .0$, $p = .995$, 95% CI [−0.19, 0.19], $BF = 0.22$) but showed a negative correlation at the phrase level ($\rho = −.7$, $p < .001$, 95% CI [−0.79, −0.57], $BF = 2.22$). The same was observed for self-esteem enhancement, with no significant correlation with control at the level of parenting dimensions ($\rho = −.05$, $p = .603$, 95% CI [−0.24, 0.14]. $BF = 0.25$) but a significant negative correlation for the parenting phrases ($\rho = −.69$, $p < .001$, 95% CI [−0.79, −0.55], $BF = 9.7 \times 10^{9}$). The associations between the phrase ratings on the dimensions are illustrated in Fig 5. Additionally, exploratory analyses revealed significant gender differences in the phrase ratings, which are reported in Supplementary Material (S2 Text).

### Discussion Study 2

Study 2 extended the findings of Study 1 by extracting the 84 most frequently mentioned parenting phrases and asking participants to rate them on the dimensions of warmth, control, and self-esteem enhancement. These ratings provided insight not only into the perceived qualities of individual phrases but also into how evaluations varied across certain

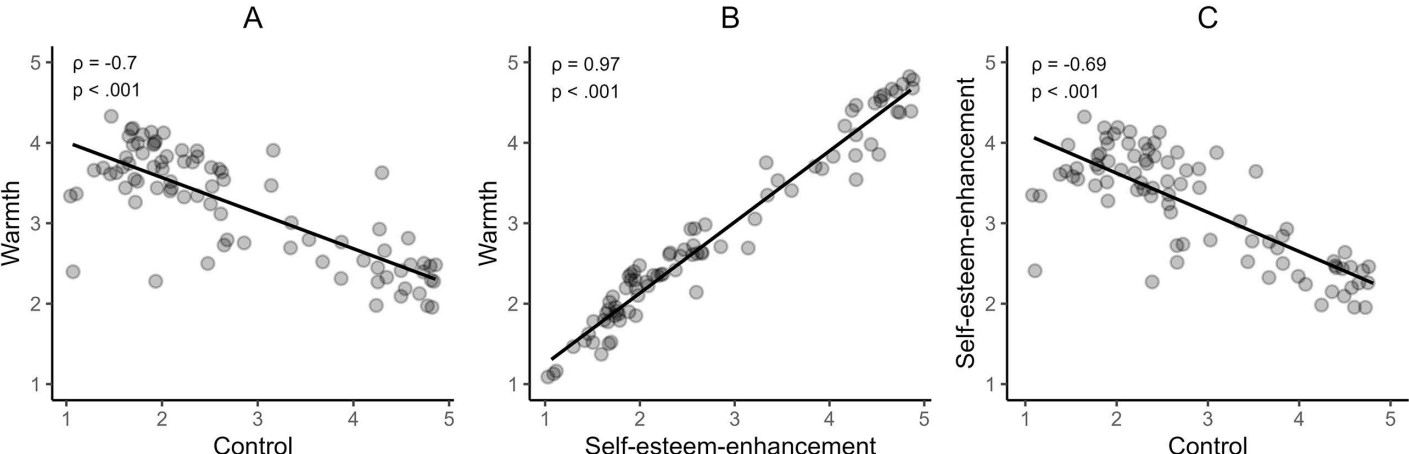

**Fig 5. Scatter plots illustrating the relationship between parenting behaviour ratings for the different dimensions.** *Spearman correlation coefficients (ρ) and p-values are indicated on each plot.*

sociodemographic characteristics. We hypothesized that participants' perceptions of their own upbringing would serve as a reference point for comparison. Specifically, participants who experienced a warmer, more controlling, or more self-esteem-enhancing upbringing would rate the phrases as lower on the corresponding dimension. While no significant associations were found for warmth or control within this sample, a noteworthy trend emerged for the self-esteem-enhancement dimension: Contrary to our hypothesis, participants with a more self-esteem-enhancing upbringing rated the phrases as significantly more self-esteem-enhancing. One possible explanation, as suggested by Krueger et al. [60], is that some participants may have a general tendency to interpret and evaluate items more positively. This optimistic bias may stem from factors unrelated to parenting, or it could be a result of positive, self-esteem-enhancing parenting. Supporting this idea, a study by Milburn et al. [61] found that individuals who received harsh parental punishment in their childhood developed a hostile attribution bias. On the other hand, experiencing self-esteem-enhancing parenting may foster a more optimistic attributional style, which could lead to more positive interpretations and ratings of the phrases.

A significant age effect was found in the evaluation of phrases for the warmth dimension, but not for control or self-esteem enhancement. Among participants in the current study, younger participants rated the phrases as less emotionally warm. This may reflect a more critical view among younger generations toward parental communication. With increased access to information [20,62] and heightened media attention to parenting practices in recent decades, younger individuals may be more likely to identify certain phrases as problematic – particularly those perceived as lacking emotional warmth. While this apparent shift in attitude may influence how parenting phrases are interpreted and judged, this study found no association between age and the actual frequency of hearing parenting phrases. This contrasts with the findings of Study 1, in which participants showed age-related differences in phrase usage frequency. Taken together, the findings of Study 2 may suggest a change in parenting attitudes, regarding the warmth dimension, whereas a corresponding behavioural change does not (yet) appear to have occurred, at least within this particular sample. Similar intention-behaviour gaps have been documented in other areas of parenting research [63,64], suggesting that although caregivers may increasingly recognize beneficial behaviours, they often struggle to implement them consistently. This discrepancy underscores the importance of effective parenting interventions that support the translation of attitudes into practice.

No differences were observed based on parenthood status in this study, indicating that parents and non-parents evaluated the parenting phrases in a similar manner. Importantly, however, a sensitivity analysis revealed that the present sample provided 80% power (α = .05) to detect effects of $d \geq .562$. Smaller effects may therefore have remained undetected.

Future research should examine potential differences between parents and non-parents using larger samples to ensure sufficient power to detect smaller effects.

Exploratory analyses of the relationships between the assessment of the different dimensions concerning participants' own upbringing revealed interesting results: Warmth and self-esteem enhancement were strongly correlated, both on the dimension level as well as on the phrase level. This suggests that participants of the current study did not clearly differentiate between these two attributes and that warm parenting was perceived as beneficial to children's self-esteem. Interestingly, while warmth and control were not significantly associated at the dimension level – consistent with their theoretical orthogonality [22] – they were significantly negatively correlated at the phrase level. This suggests that individual parenting phrases tend to be perceived as either warm or controlling, but not both. A similar pattern emerged for control and self-esteem enhancement, which were also uncorrelated at the parenting style dimension level but were significantly negatively correlated at the phrase level. These explorative findings can be interpreted through the lens of Relational Framing Theory [14], which posits that relational messages are interpreted in ways that emphasize one relational quality (either affiliation-disaffiliation or dominance-submissiveness) at the expense of the other. In the context of parenting phrases, this would imply that perceiving an expression as relevant to control may inhibit its interpretation as relevant to warmth, and vice versa. These findings suggest a distinctive dynamic in how parenting phrases function as verbal expressions of parenting behaviour, highlighting the valuable contribution of communication theories in complementing traditional parenting frameworks and deepening our understanding of these forms of caregiver-child interaction.

Taken together, Study 2 offered a more methodologically robust approach to categorising parenting phrases than the initial classification in Study 1. It was also the first to shed light on sociodemographic differences in perceptions of this specific form of parental communication, revealing how factors such as experience with parenting and age may shape how parenting phrases are interpreted. Additionally, the findings offer initial evidence that communication theories may enrich our understanding of how parenting messages are conveyed and received, laying a foundation for future research.

## General discussion

### Summary and interpretation of results

This set of two studies aimed to explore the new topic of parenting phrases as a distinct expression of parental verbal behaviour within a German-speaking sample. The findings demonstrate that caregivers can employ a wide range of specific phrases when interacting with their children. In Study 1, a total of 309 parenting phrases were collected, of which 84 were rated by participants in Study 2.

In exploring how the phenomenon of parenting phrases can be situated within existing theoretical frameworks, explorative findings from both studies suggest that these phrases may be effectively interpreted through a lens of Relational Framing Theory by McLaren [14]. According to this theory, relational messages are interpreted through either an affiliation-disaffiliation frame or a dominance-submissiveness frame. The observed pattern of phrase ratings, where phrases were often perceived as either warm or controlling, but rarely both, is consistent with such a framing account. Importantly, however, these analyses were exploratory and were not designed as confirmatory tests of Relational Framing Theory. Rather, the findings generate theory-informed hypotheses that warrant more direct examination in future research. Compared to traditional parenting style theories, such as those proposed by Baumrind [16] and Maccoby & Martin [17], Relational Framing Theory specifically addresses how verbal expressions are interpreted. By doing so, it may capture nuances overlooked by broader behavioural frameworks and offer a valuable addition for understanding the dynamics of verbal interaction in the parenting context.

While this theoretical perspective deepens our understanding of how the content of parental verbal expressions may be interpreted, an equally important question concerns how the use of such language persists and changes within families and broader societal contexts. To address this, the studies examined potential continuities and discontinuities in the use of the phrases across generations. Results from Study 1 indicate that, in this sample, participants remembered more negative and fewer positive phrases from their childhood than they reported using towards their own children. This pattern may

suggest that caregivers are making a conscious effort to use more intentional language than their parents did. However, it remains unclear whether this discrepancy results from an actual change in phrase usage or from a bias in favour of participants' own parenting practices. No evidence was found for a relationship between the frequency of positive and negative parenting phrases heard by participants' caregivers and those used by participants' towards their own children. In other words, in the current study, participants who reported hearing more positive or negative phrases from their caregivers did not report using more of these phrases themselves. This finding contrasts with previous research demonstrating an intergenerational transmission of general parenting behaviour [25]. The results suggest that the socialisation of parenting phrases may follow a different pathway than the transmission of broader parenting behaviour, raising important questions about the mechanisms that contribute to the use of specific verbal expressions in parenting.

Beyond intergenerational transmission within families, both Study 1 and 2 examined age-related differences in parenting phrase usage as a means of identifying possible historical trends in caregiver-child communication. While Study 1 found that older participants reported using significantly more negative phrases towards their children and hearing significantly fewer positive phrases, Study 2 found no significant age differences for the experience of parenting phrases. The only significant age-related effect observed in Study 2 concerned the rating of phrases on the warmth dimension, with younger participants rating the phrases as significantly less warm. Overall, the results from both studies offer only limited evidence for a shift in parenting behaviour. Instead, they point more toward subtle changes in the perception of the phrases (Study 2), with only a few indications of changes in their actual use (Study 1). Further research is needed to clarify whether, and in what ways, parenting phrase interpretation and usage have genuinely changed over time.

To fully understand the role of parenting phrases, it is crucial to move beyond depictions of general usage patterns and consider who is interpreting these messages. Communication is not a one-way process, but requires the interpretative action of the recipient. Individual characteristics of the recipient could shape how messages are understood and, consequently, how they impact the listener. Study 2 examined interpersonal differences in the rating of parenting phrases in relation to specific socio-demographic factors: In this study, participants' childhood experiences of parenting, as well as their age, significantly related to their ratings. Specifically, those who reported having experienced more self-esteem-enhancing parenting in childhood were more likely to perceive the phrases as self-esteem-enhancing. Younger participants tended to rate the phrases as less warm. No association between parenthood and the ratings of the phrases was found in this sample, indicating that parents and non-parents perceived the phrases similarly. This initial analysis of factors associated with the interpretation of parenting phrases provides a first insight into the complexity of how such messages are perceived. Future research could investigate additional variables that could be related to individuals' interpretations. By examining how different groups may perceive these statements in varying ways, we can deepen our understanding of how parenting phrases – and their subjective interpretation – may affect the individual they are directed toward.

Repeated parental messages may form a child's core beliefs and thereby exert a significant influence on their development. As a first step toward understanding the developmental impact of parenting phrases, Study 1 investigated the association between phrases recalled from childhood and participants' self-esteem. The literature suggests a significant association between parenting behaviour and self-esteem [26–35]. Surprisingly, the frequency with which participants reported hearing phrases categorised as self-esteem-enhancing or self-esteem-inhibiting did not correlate with their actual self-esteem in the current sample. (A detailed discussion of potential conceptual and methodological explanations is provided in the Discussion of Study 1.) This unexpected finding raises questions about the conceptual validity of these phrase categories. Even though certain expressions may intuitively appear to support or hinder self-esteem, research has repeatedly demonstrated that such intuitions can be misleading. For instance, adults often use forms of praise which they believe to be beneficial for children's self-esteem, when in fact, these can have unintended negative effects [65–67]. Supporting this concern, in Study 2, ratings of warmth were positively correlated with self-esteem enhancement at both the behavioural and parenting phrase levels. The particularly strong correlation observed at the phrase level ($r = .98$) suggests that participants may have perceived these two constructs as largely overlapping rather than distinct. These findings

highlight the need for refining the operational definitions and measurement approaches for these constructs in future research. Although no direct association was found between recalled parenting phrases and self-esteem, further investigation remains important. Parenting phrases represent a complex form of verbal interaction that may subtly influence children's development. Previous research has reported associations between parental messages and children's worldview [2] and aggressive behaviour [7]. Future research should further investigate the relationship between parenting phrases and developmental outcomes. Such insights could guide targeted interventions, supporting caregivers in more deliberately using a language that fosters children's well-being and positive development.

## Limitations

While the present studies provide valuable insights into the evaluation and use of parenting phrases, several limitations warrant consideration. First, in Study 1 some categories demonstrated moderate-to-low interrater agreement. Especially the "low control" category was characterized by substantial uncertainty in the ratings and comprised only a small number of items. This may reflect its conceptual ambiguity as an absence of directive behaviour, which is more difficult to infer from isolated phrases and may overlap with autonomy-supportive or warm expressions. Accordingly, conclusions regarding this category should be interpreted with caution. Study 2 addressed this limitation by adopting a dimensional evaluation approach, representing a first step toward a more robust assessment of parenting phrases. Future research could also address this issue by refining the definition of low control and providing more contextual information to facilitate clearer differentiation from related constructs such as autonomy support.

Another concern relates to the interpretations of hypotheses involving the parent subsamples, many of which yielded null results. Sensitivity analyses indicated sufficient statistical power to detect moderate-to-large effects in both studies. To address the challenge of detecting smaller effects, we used Bayesian approaches when sub-samples were considered. As both studies relied on convenience samples with only a small proportion of male participants, the generalizability of the findings is limited. Future studies should aim to recruit more balanced samples, including a larger proportion of parents, to enable more robust and generalizable conclusions.

Lastly, the retrospective and self-report survey method employed in the present studies also presents certain challenges. Parenting behaviour may be a topic susceptible to social desirability responses, potentially influencing participants' reports of both their own and their caregivers' behaviour. Moreover, recollections of childhood experiences may be biased and shaped by reinterpretation processes. Hardt and Rutter [68] demonstrated that "reports of experiences that rely heavily on judgement or interpretation" (p. 260) are particularly susceptible to bias. Given the interpretative nature of verbal communication, this is likely to apply to participants recall of parenting phrases as well. Nonetheless, perceived parenting, in particular, appears to be a significant predictor of key psychological and clinical outcomes [69,70]. Although it may not always align with objective parenting behaviour, perceived and recalled parenting remains highly valuable in psychological research. In addition, future research could address these limitations by employing observational and longitudinal designs, which would allow for a more direct examination of parenting communication over time.

## Implications and future research directions

The current research project is the first to provide insights into the complex topic of parenting phrases as one important aspect of parental communication within a German-speaking population. It offers valuable initial findings on the use and interpretation of these phrases, laying a foundation for future research. However, this work represents only a starting point with many questions remaining unanswered. For instance, little is known about the contexts in which these phrases are used, the interpretative processes that shape their intended and perceived meaning, or the specific developmental outcomes that might be associated with them. Further research may contribute to a more elaborate understanding of the mechanisms and influencing factors that contribute to the use of parenting phrases, as well as a better, more nuanced

knowledge of the potential effects they can have on children's development. Communication theories should be taken into consideration, as they offer a deeper understanding of parenting phrases, extending beyond the insights provided by traditional parenting theories. The current project provides a collection of parenting phrases commonly used in Germany that is available to be used in future research. They can serve as a valuable stimulus set in German-speaking samples, and for international comparisons of parental language use, offering multiple avenues for future investigations. Potential applications include the development of phrase-based parenting measures or using the collection as a coding framework for observational studies. In applied contexts, the corpus may also inform intervention studies aimed at promoting supportive parental language.

In conclusion, this research project draws attention to an underexplored aspect of parenting, aiming to not only advance scientific understanding of family interactions but also to encourage broader reflection within communities on the role of language in caregiver-child relationships.

## Supporting information

**S1 Table. Parenting phrases presented in Study 1 (64 phrases) and their assigned categories.**
(XLSX)

**S2 Table. Parenting phrases collected in Study 1 (309 phrases).**
(XLSX)

**S3 Table. Parenting phrases presented in Study 2 (84 phrases) with their mean ratings and standard deviations.**
(XLSX)

**S1 Text. Correction of the preregistration of Study 1.**
(PDF)

**S2 Text. Exploratory analyses of gender differences in parenting phrases ratings.**
(PDF)

**S1 Analysis. Prerequisite analyses Study 1 and results with and without outliers.**
(PDF)

**S2 Analysis. Prerequisite analyses Study 2 and results with and without outliers.**
(PDF)

## Author contributions

**Conceptualization:** Erika Kljucak, Nourat N. Alazza, Pia Hemme, Louisa Kulke.

**Data curation:** Erika Kljucak.

**Formal analysis:** Erika Kljucak.

**Funding acquisition:** Louisa Kulke.

**Investigation:** Erika Kljucak, Nourat N. Alazza, Pia Hemme.

**Resources:** Louisa Kulke.

**Supervision:** Louisa Kulke.

**Writing – original draft:** Erika Kljucak.

**Writing – review & editing:** Erika Kljucak, Nourat N. Alazza, Pia Hemme, Louisa Kulke.

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
