## [Decision Letter · Decision Letter 0]

29 Dec 2025

Dear Dr. Kulke,

Thank you for submitting your manuscript to PLOS ONE. After careful consideration, we feel that it has merit but does not fully meet PLOS ONE’s publication criteria as it currently stands. Therefore, we invite you to submit a revised version of the manuscript that addresses the points raised during the review process.

We look forward to receiving your revised manuscript.

Kind regards,

Henri Tilga, PhD

Academic Editor

PLOS One

Journal Requirements:

Reviewers' comments:

Reviewer's Responses to Questions

**Comments to the Author**

1. Is the manuscript technically sound, and do the data support the conclusions?

Reviewer #1: Yes

Reviewer #2: Yes

2. Has the statistical analysis been performed appropriately and rigorously?

Reviewer #1: Yes

Reviewer #2: Yes

3. Have the authors made all data underlying the findings in their manuscript fully available?

Reviewer #1: Yes

Reviewer #2: Yes

4. Is the manuscript presented in an intelligible fashion and written in standard English?

Reviewer #1: Yes

Reviewer #2: Yes

Reviewer #1: Thank you for the opportunity to review this manuscript. Overall, the article makes a meaningful and novel contribution to the study of parental communication by systematically collecting and evaluating parenting phrases in German-speaking samples. The two-study design, preregistration, and transparent reporting of statistical decisions demonstrate strong methodological rigor.

Some conceptual categories, especially low control, show limited interrater reliability and should either be better justified or more clearly presented as a limitation. The many nonsignificant associations, particularly with self-esteem and intergenerational continuity, would benefit from a more developed discussion of possible methodological and conceptual reasons. Finally, a brief, focused section on how future work could use this phrase corpus, together with minor edits to shorten long sentences and clarify a few tables, would further strengthen the paper’s clarity and impact.

Reviewer #2: I would like to thank for the opportunity to review this manuscript. Please see the following comments to consider to further increase the quality of this manuscript.

This manuscript addresses an original and underexplored aspect of parenting research: the systematic collection, categorization, and evaluation of commonly used parenting phrases. The two-study design is well-motivated, the integration of qualitative and quantitative approaches is appropriate, and the attempt to bridge parenting style theory with Relational Framing Theory (RFT) is innovative. The work offers a valuable descriptive foundation and opens a promising research avenue in parental communication.

However, the manuscript would benefit from clearer conceptual positioning, stronger justification of analytic decisions, more cautious interpretation of null findings, and greater transparency regarding methodological limitations, particularly concerning sample size, measurement validity, and retrospective reporting. Several sections would also profit from tighter structure and more explicit linking of hypotheses, results, and theory.

With targeted revisions, this manuscript could make a solid contribution to the literature on parenting and communication.

Comments:

While the manuscript defines parenting phrases broadly, the construct remains conceptually diffuse. Parenting phrases overlap with related constructs such as parental messages, verbal discipline, emotional socialization, and psychological control, yet these distinctions are not sufficiently clarified.

The application of Relational Framing Theory (RFT) is intriguing but remains largely post hoc and underdeveloped, particularly in Study 1.

Both studies rely on convenience samples with a strong gender imbalance and a relatively small subsample of parents, which limits the robustness of intergenerational and parenting-use conclusions.

Avoid phrasing that implies population-level conclusions (e.g., “participants reported using fewer negative phrases”) and replace with sample-bound interpretations.

Consider adding a power sensitivity analysis or explicitly stating the smallest detectable effects.

The reliance on retrospective self-report for childhood experiences raises substantial concerns about memory bias, reinterpretation, and social desirability, particularly given the evaluative nature of the phrases.

The categorization of phrases shows uneven interrater reliability, especially for the “low control” category, which also contains very few items.

A large proportion of hypotheses yielded null results, yet interpretation occasionally implies theoretical support through absence of effects.

The use of global self-esteem may be too distal and broad to capture the effects of specific parenting phrases.

Please suggest alternative or complementary outcome measures for future studies (e.g., contingent self-esteem, internalized criticism, attachment security). For example, please see a recent study by Meerits et al., (2023) in which need-supportive parenting program was designed and its effectiveness tested. Perhaps it would be interesting suggestion for future research to test motivational parenting program in future research to test its effectiveness to achieve desired outcomes.

Meerits, P.-R., Tilga, H., & Koka, A. (2023). Web-based need-supportive parenting program to promote physical activity in secondary school students: A randomized controlled pilot trial. BMC Public Health, 23(1), 1627. https://doi.org/10.1186/s12889-023-16528-4

**Do you want your identity to be public for this peer review?** For information about this choice, including consent withdrawal, please see our For information about this choice, including consent withdrawal, please see our Privacy Policy .

Reviewer #1: **Yes:** Claudiu ComanClaudiu Coman

Reviewer #2: No

---

## [Author Response · Author response to Decision Letter 1]

2 Mar 2026

We thank the editor and the reviewers for their careful evaluation and their helpful comments, which we feel have substantially improved the manuscript. We have addressed all comments as detailed below and hope that you now find the manuscript suitable for publication.

Reviewer 1:

Comment 1:

Some conceptual categories, especially low control, show limited interrater reliability and should either be better justified or more clearly presented as a limitation.

Response:

We agree that the limited interrater reliability observed for some of the categories warrants clearer discussion. Accordingly, we have revised the Discussion section of Study 1 and added a “Limitations” section to the General Discussion to more explicitly acknowledge this limitation. (p. 20, lines 444-454; p. 40, lines 941-951).

Comment 2:

The many nonsignificant associations, particularly with self-esteem and intergenerational continuity, would benefit from a more developed discussion of possible methodological and conceptual reasons.

Response:

We agree that the pattern of nonsignificant associations is interesting for further discussion. Accordingly, we have expanded the Discussion sections to more thoroughly address potential methodological and conceptual explanations. We added a sensitivity analysis to interpret the absence of significance in the context of the studies statistical power (p. 21, lines 460-474; pp. 23, lines 530-559). We also clarify how these considerations may inform the interpretation of the absent findings and guide future research. Particularly, we related our findings to the Relational Framing Theory (pp. 427-440).

Comment 3:

Finally, a brief, focused section on how future work could use this phrase corpus, together with minor edits to shorten long sentences and clarify a few tables, would further strengthen the paper’s clarity and impact.

Response:

We have followed this suggestion and made several revisions to improve clarity. Specifically, we added a brief, focused paragraph in the Discussion outlining potential uses of the phrase corpus in future research (p. 42, lines 986-993). In addition, we edited several long sentences throughout the manuscript for conciseness and clarity and revised Table 1 and Tables 3-5 to improve readability.

Reviewer 2:

Comment 1:

However, the manuscript would benefit from clearer conceptual positioning, stronger justification of analytic decisions, more cautious interpretation of null findings, and greater transparency regarding methodological limitations, particularly concerning sample size, measurement validity, and retrospective reporting.

Response:

We thank the reviewer for their thorough evaluation and for highlighting areas for improvement. We have carefully revised the manuscript to clarify conceptual positioning, justify analytic decisions, interpret null findings cautiously, and transparently discuss methodological limitations, as detailed in the responses below.

Comment 2:

Several sections would also profit from tighter structure and more explicit linking of hypotheses, results, and theory.

Response:

We have revised several sections of the manuscript to improve structural clarity and to more explicitly link hypotheses, results, and theoretical considerations. Specifically, we clarified the connections between the predicted outcomes, the observed results and their interpretation in the Results and Discussion sections. These changes aim to make the theoretical rationale and empirical findings more coherent and easier to follow.

Comment 3:

While the manuscript defines parenting phrases broadly, the construct remains conceptually diffuse. Parenting phrases overlap with related constructs such as parental messages, verbal discipline, emotional socialization, and psychological control, yet these distinctions are not sufficiently clarified.

Response:

Thank you for making these relevant connections. We have revised the Introduction section to more clearly delineate the construct of parenting phrases and its relation to related constructs, including parental messages, verbal discipline, verbal aggression, psychological control and emotional socialization. Specifically, we now explain, that the construct of parenting phrases encompasses the linguistic component embedded within broader parenting behaviour (pp. 3-4, lines 57-81). These clarifications aim to reduce conceptual ambiguity and provide readers with a clearer understanding of the theoretical framework guiding our study.

Comment 4:

The application of Relational Framing Theory (RFT) is intriguing but remains largely post hoc and underdeveloped, particularly in Study 1.

Response:

We have revised the Discussion section of Study 1 and the General Discussion to more clearly integrate Relational Framing Theory (RFT) and to more cautiously communicate its exploratory role in the present study. Specifically, we now provide a clearer rationale for how RFT informs the interpretation of the parenting phrases, and explicitly note that the application of RFT is exploratory and intended to guide future hypothesis development (pp. 19-20, lines 422-440; p. 37, lines 852-865).

Comment 5:

Both studies rely on convenience samples with a strong gender imbalance and a relatively small subsample of parents, which limits the robustness of intergenerational and parenting-use conclusions.

Response:

We have revised the General Discussion section to more explicitly acknowledge the reliance on convenience samples, the gender imbalance, and the relatively small number of parents in our samples. We also discuss how these factors may limit the generalizability of the current findings and suggest that future research should aim to recruit larger, more representative samples (pp. 40-41, lines 952-959).

Comment 6:

Avoid phrasing that implies population-level conclusions (e.g., “participants reported using fewer negative phrases”) and replace with sample-bound interpretations.

Response:

We have revised the manuscript throughout to ensure that all conclusions are framed as sample-bound rather than population-level. These revisions appear throughout the Discussion sections and aim to accurately reflect the scope of inference supported by our data.

Comment 7:

Consider adding a power sensitivity analysis or explicitly stating the smallest detectable effects.

Response:

We have added a brief power sensitivity analysis, reporting the smallest effect sizes that could be reliably detected given our sample sizes in each study. This information clarifies the statistical power of our analyses and helps contextualize the nonsignificant associations (p. 21, lines 469-474; p. 23, lines 530-535; p. 35, lines 813-817).

Comment 8:

The reliance on retrospective self-report for childhood experiences raises substantial concerns about memory bias, reinterpretation, and social desirability, particularly given the evaluative nature of the phrases.

Response:

Thank you for pointing out the confounds of retrospective self-report. We agree with you that these may play a role and have revised the Discussion section to explicitly address the potential influence of retrospective self-report, including memory bias, reinterpretation, and social desirability effects, particularly in relation to the evaluative nature of parenting phrases. We suggest that futures studies could complement self-report data with observational or longitudinal designs to mitigate these concerns (p. 41, lines 960-972)

Comment 9:

The categorization of phrases shows uneven interrater reliability, especially for the “low control” category, which also contains very few items.

Response:

We agree that the limited interrater reliability observed for some of the categories, as well as the small number of items categorized as “low control”, warrants clearer discussion. Accordingly, we have revised the Discussion section of Study 1 as well as the General Discussion to more explicitly acknowledge these limitations. (p. 20, lines 444-454; p. 40, lines 941-951).

Comment 10:

A large proportion of hypotheses yielded null results, yet interpretation occasionally implies theoretical support through absence of effects.

Response:

We have made several revisions throughout the Discussion sections to ensure that null findings are interpreted cautiously within the methodological and conceptual boundaries of the studies, making sure that they are not presented as evidence supporting theoretical predictions. We have furthermore provided Bayes Factors for Null effects, comparing the likelihood of the Null hypothesis with the alternative hypothesis to get an estimate of the likelihood of Null effects.

Comment 11:

The use of global self-esteem may be too distal and broad to capture the effects of specific parenting phrases.

Please suggest alternative or complementary outcome measures for future studies (e.g., contingent self-esteem, internalized criticism, attachment security). For example, please see a recent study by Meerits et al., (2023) in which need-supportive parenting program was designed and its effectiveness tested. Perhaps it would be interesting suggestion for future research to test motivational parenting program in future research to test its effectiveness to achieve desired outcomes.

Response:

We agree that global self-esteem may represent a relatively distal outcome for capturing the effects of specific parenting phrases. We have therefore expanded the Discussion of Study 1 to suggest more proximal outcome measures for future research and to note that the phrase corpus could also be used in intervention-based designs to examine whether changes in parental language are associated with changes in relevant outcome variables (p. 24, lines 543-559).

We would like to thank the reviewers again for their valuable feedback which we believe has helped to thoroughly improve the manuscript. We hope that you now find it suitable for publication.

---

## [Decision Letter · Decision Letter 1]

23 Mar 2026

"Because I said so." - Collection and evaluation of parenting phrases in German-speaking samples

PONE-D-25-60771R1

Dear Dr. Kulke,

We’re pleased to inform you that your manuscript has been judged scientifically suitable for publication and will be formally accepted for publication once it meets all outstanding technical requirements.

Kind regards,

Henri Tilga, PhD

Academic Editor

PLOS One

Additional Editor Comments (optional):

Reviewers' comments:

Reviewer's Responses to Questions

**Comments to the Author**

Reviewer #2: All comments have been addressed

2. Is the manuscript technically sound, and do the data support the conclusions?

Reviewer #2: Yes

3. Has the statistical analysis been performed appropriately and rigorously?

Reviewer #2: Yes

4. Have the authors made all data underlying the findings in their manuscript fully available?

Reviewer #2: Yes

5. Is the manuscript presented in an intelligible fashion and written in standard English?

Reviewer #2: Yes

Reviewer #2: Authors have done well job on revising their manuscript. I think this manuscript is ready to be published.

**Do you want your identity to be public for this peer review?** For information about this choice, including consent withdrawal, please see our For information about this choice, including consent withdrawal, please see our Privacy Policy .

Reviewer #2: No

---

## [Editor Report · Acceptance letter]

PONE-D-25-60771R1

PLOS One

Dear Dr. Kulke,

I'm pleased to inform you that your manuscript has been deemed suitable for publication in PLOS One. Congratulations! Your manuscript is now being handed over to our production team.

Kind regards,

on behalf of

Dr. Henri Tilga

Academic Editor

PLOS One